# Observed changes in the temperature and height of the globally resolved lapserate tropopause

Florian Ladstädter<sup>1</sup>, Matthias Stocker<sup>1</sup>, Sebastian Scher<sup>1</sup>, and Andrea K. Steiner<sup>1</sup>

<sup>1</sup>Wegener Center for Climate and Global Change (WEGC), University of Graz, Graz, Austria

**Correspondence:** Florian Ladstädter (florian.ladstaedter@uni-graz.at)

Abstract. The tropopause is a key indicator of atmospheric climate change, influenced by both the troposphere and stratosphere. Here we present a global view of tropopause changes, using high-resolution GNSS radio occultation data from 2002 to 2024. We identify significant trends in lapse rate tropopause (LRT) temperature and height with seasonal and regional detail. The tropical LRT has warmed, with particularly strong warming (> 1 K/decade) over the South Pacific during austral spring and summer, while height changes remain largely insignificant. Outside the tropics, LRT temperature changes are confined to southern high latitudes in winter, showing cooling of up to 1 K/decade. Notably, LRT height has increased significantly across most extratropical regions, with localized trends exceeding 200 m/decade over Asia and the Middle East during Northern Hemisphere winter. An exception is the LRT height decreases over the South Pacific, coinciding with a LRT warming in that region. These results highlight the interrelated effects of tropospheric and stratospheric changes and demonstrate the value of precise tropopause monitoring for detecting ongoing changes in the global climate system.

### 1 Introduction

The tropopause, acting as transition zone between the troposphere and the stratosphere, is a sensitive indicator of changes in the atmospheric climate system. In a simple conceptual model, a rise of the tropopause can be caused by either upper-tropospheric warming, e.g. due to climate change and corresponding increase in surface temperature, or lower-stratospheric cooling, e.g. due to an increase in atmospheric  $CO_2$  or ozone depletion (Santer et al., 2003b, 2004). This relation has been confirmed in several observational (e.g., Seidel and Randel, 2006) and modeling studies (e.g., Son et al., 2009).

The tropopause is strongly coupled to stratospheric temperature trends, meaning that radiative or dynamical changes in the stratosphere strongly affect the tropopause (Shepherd, 2002; Seidel and Randel, 2006; Birner, 2010a). Consequently, the ozone recovery of the last decades along with its impact on lower stratospheric temperatures, and the potential long-term trend in the residual meridional circulation in the stratosphere (Brewer-Dobson circulation (BDC)) are also expected to change the tropopause structure (Birner, 2010a; Weber et al., 2011; Butchart, 2014).

A potential poleward shift of the subtropical jet streams, affecting storm tracks and hurricane formation, will be reflected in subtropical tropopause height trends (Seidel and Randel, 2007; Lorenz and DeWeaver, 2007; Birner, 2010b; Meng et al., 2021).

In the tropics, the structure of the tropical tropopause layer, and specifically the temperature of the cold point tropopause there, is decisive for the transition of tropospheric air into the stratosphere, and sets the level of stratospheric water vapor concentration (Randel and Jensen, 2013).

Numerous studies have discussed variability and changes of the tropopause, focusing e.g. on the tropopause as indicator of climate change (Sausen and Santer, 2003; Santer et al., 2004), on chemistry and dynamics of the upper troposphere-lower stratosphere region (Holton et al., 1995; Son et al., 2009; Birner, 2010a; Gettelman et al., 2011), on the arctic tropopause (Highwood et al., 2000), on the tropical tropopause layer (Randel and Jensen, 2013; Randel and Wu, 2015; Wang et al., 2016; Lin et al., 2017), on double tropopause structure (Peevey et al., 2014; Wilhelmsen et al., 2020), and on observed changes of the tropopause (Seidel et al., 2001; Santer et al., 2003a; Seidel and Randel, 2006; Schmidt et al., 2008; Rieckh et al., 2014; Xian and Homeyer, 2019; Meng et al., 2021).

Detailed monitoring of tropopause changes is particularly important, because many atmospheric changes are reflected in characteristic changes in the structure of the tropopause. Recent work on the observed changes of the tropopause have focused on the tropics (Zolghadrshojaee et al., 2024), the Northern Hemisphere (Meng et al., 2021), or long-term radiosonde trends (Xian and Homeyer, 2019), mostly focusing in tropopause height changes. Here, we aim to go beyond previous studies by providing both a global and a regional view of how the tropopause temperature and height have changed over the past 23 years, using Global Navigation Satellite System (GNSS) radio occultation (RO) observations.

35

50

Among the various data sets available for the temperature of the upper air atmosphere, GNSS RO is particularly well suited to monitor the parameters of the tropopause for the following reasons: (a) GNSS RO is available globally with an adequate horizontal sampling density; (b) in the upper troposphere and lower stratosphere (UTLS), where the tropopause is located, GNSS RO is almost entirely pure observational data, i.e. background information from forecast models do not affect the quality of the observational information in this region (Steiner et al., 2020a); (c) GNSS RO has a high vertical resolution of approximately 100 m within the UTLS, enabling to resolve the tropopause structure in detail (Zeng et al., 2019); and (d), over 20 years of continuous data are now available. Additionally, the long-term stability of GNSS RO, based on phase change measurements with the precision of atomic clocks (Steiner et al., 2011), underscores our confidence in the trend estimates derived from the GNSS RO time series.

The location and structure of the tropopause region can be determined with a range of different metrics, such as the coldpoint tropopause often used in the tropics (Highwood and Hoskins, 1998), the thermal tropopause (WMO, 1957), the dynamical definition based on potential vorticity gradients, or the chemical tropopause. See, Gettelman et al. (e.g., 2011) for a short review on different tropopause definitions. Among these definitions, the thermal tropopause based on the standard definition from the World Meteorological Organization (WMO) has the advantage of being easily computed from temperature profiles alone, and of being adequately defined in almost all atmospheric states and latitudes. We therefore choose the thermal, lapse rate tropopause (LRT) definition consistently for all global regions.

# 2 Data and Methods

75

In order to investigate changes in tropopause parameters, we utilize GNSS RO data from 2002 to 2024 to acquire high-resolution temperature profiles. The GNSS RO method involves the transmission of electromagnetic signals emitted by GNSS satellites, and received by low Earth orbit satellites. These signals are used, in conjunction with precise orbit information, to determine the bending of the signal due to the refractive properties of the atmosphere (Kursinski et al., 1997). Assuming water vapor can be neglected, refractivity is directly related to temperature and pressure, as described by the Smith-Weintraub formula (Smith and Weintraub, 1953). Utilizing this formula together with the hydrostatic equation yields profiles of atmospheric temperature and pressure. In lower tropospheric regions where water vapor cannot be neglected, additional (background) information is necessary for the retrieval. In the context of our study, where we compute lapse rate tropopause parameters from temperature profiles, we can safely assume that water vapor is negligible.

The high quality and good vertical resolution of GNSS RO, particularly in the UTLS region (Gorbunov et al., 2004), offer a unique opportunity to investigate the globally resolved tropopause structure in detail. The effective vertical resolution of GNSS RO around the tropopause is about 100 m to 200 m (Zeng et al., 2019). The long-term stability and low structural uncertainty (Steiner et al., 2020a) are favorable for reliable trend detection, even considering the climatologically rather short time period of 23 years used in this study. The systematic uncertainty in the UTLS is estimated to be below 0.1 K, with the lowest values around the tropopause (Scherllin-Pirscher et al., 2011b, 2021; Schwarz et al., 2017). Random uncertainty can already be neglected for small aggregation sizes, and the structural uncertainty due to processing choices is less than 0.1 K in that region (Steiner et al., 2020b).

We employ GNSS RO temperature profiles processed by the Radio Occultation Meteorology Satellite Application Facility (ROM SAF) (Gleisner et al., 2020). We apply the tropopause algorithm to the dry temperature profiles in order to avoid the impact of background information that is inherent in the physical temperature retrieval.

To compute the tropopause temperature and height, we follow the lapse rate tropopause algorithm defined by the WMO (WMO, 1957) for each individual RO dry temperature profile. While other tropopause definitions exist, each of which possesses its own set of advantages and disadvantages, we consider the LRT algorithm to be the best compromise when analyzing the tropopause on a global scale. It is commonly used, facilitates comparisons with other studies, is straightforwardly computable on temperature profiles, and does not require additional parameters. The WMO defines a LRT as "the lowest level at which the lapse rate decreases to 2 °C/km or less, provided also the average lapse rate between this level and all higher levels within 2 km does not exceed 2 °C/km." As the minimum tropopause height we employ an empirical limit as defined by Son et al. (2011): 7.5 km + 2.5 km  $\cdot$ cos(2 $\phi$ ), where  $\phi$  is the latitude. This limit is employed to prevent erroneous interpretation of dry temperature profile gradients as a tropopause by the lapse rate algorithm, which can occur in the presence of higher water vapor content in the lower troposphere. According to the WMO lapse rate definition, a second tropopause might occur above the first, which is a phenomenon common in the extratropics (e.g., Randel et al., 2007). In this study, however, only the lowest (first) tropopause is considered when multiple tropopauses are found in the temperature profile.

The resulting LRT values are aggregated to monthly means at two spatial resolutions: zonal means with  $5^{\circ}$  latitudinal resolution, and  $10^{\circ} \times 10^{\circ}$  gridded, the latter to investigate regionally resolved trend patterns. From the resulting monthly mean fields, anomalies are created by subtracting from each month the mean monthly climatology over 2007 to 2021, effectively removing the seasonal cycle.

We note that the number of measurements involved in the RO time series varies due to the transition from the early single-mission RO period (2002 to mid-2006) with less measurements to the later period (mid-2006 to 2024) when more RO missions contributed (Gleisner et al., 2020). In many applications, it is favorable to apply a correction for the resulting sampling bias caused by incomplete atmospheric sampling using a background field such as the ERA5 reanalysis (Foelsche et al., 2008; Scherllin-Pirscher et al., 2011a). In our case, applying such a correction does not alter the overall trend values and patterns (not shown). Therefore, we have chosen not to correct for these sampling inhomogeneities here, since the decreased vertical resolution of such a background field could tamper with the pure observational information contained in the RO profiles.

Trends are calculated using multiple linear regression. We account for autocorrelation by correcting the effective degrees of freedom (Santer et al., 2000). El Niño—Southern Oscillation (ENSO) and Quasi-Biennial Oscillation (QBO), two of the most prominent internal modes of variability in the troposphere and stratosphere, are accounted for in the regression. We use monthly sea surface temperature anomalies of the Niño 3.4 region lagged by three months as an index for ENSO. We use the leading three components of a principal component analysis over all available pressure levels of the Singapore wind data as an index for QBO, though the QBO regressor has only a small impact on the trend results in the tropopause region.

### 3 Results

Figure 1 illustrates the zonal mean LRT temperature and height trends for the time span between 2002 to 2024, based on monthly anomalies, as a function of latitude. We observe a significant increase of the tropical LRT temperature of approximately 0.3 K/decade (Fig. 1 left column). This increase is not symmetrical around the equator, but larger in the Southern hemisphere (SH), reaching a maximum at SH midlatitudes with approximately 0.5 K/decade. In the Northern hemisphere (NH), outside the tropics, trends are generally not significant at the 95% level and close to zero at midlatitudes. Polar regions show negative LRT temperature trends, especially in the SH high latitudes, but the negative trends are significant only close to the south pole.

In contrast to temperature, the height of the tropical zonal-mean LRT level has remained virtually unchanged during the last two decades. This is somehow surprising, because rising tropospheric and decreasing stratospheric temperatures should be correlated to a rising LRT (Santer et al., 2004). However, recent studies have indicated that the lowermost tropical stratosphere has not further cooled (Steiner et al., 2020a), but rather has warmed during that time period (Ladstädter et al., 2023). The absence of observed height increases may be related to this warming.

The LRT is rising in both hemispheres significantly outside the tropics, with about 50 m/decade in the NH midlatitudes and up to 100 to 200 m/decade in high latitudes. This aligns with Meng et al. (2021), who investigated tropopause height changes

**Figure 1.** Observed trends in lapse rate tropopause temperature (top row) and altitude (bottom row), from 2002 to 2024, with 5° latitudinal resolution. The left column shows the trends over all included months, the second through fifth columns show seasonal trends for DJF, MAM, JJA, and SON. The bars indicate the 95% confidence level, and trend values significant at the 95% level are marked with an asterisk.

for the NH only, and found an increase of about 50 m/decade to 60 m/decade between  $20^{\circ}$  N and  $80^{\circ}$  N over both investigated time periods (1980 to 2000 and 2001 to 2020).

As with LRT temperature trends, LRT height trends reveal an asymmetry around the equator. In the SH subtropics, LRT height exhibits even a slight decrease, although this decrease is not statistically significant at the 95% level. This phenomenon coincides with a prominent warming of the SH lower stratosphere at these latitudes (Ladstädter et al., 2023).

A seasonal examination of the zonal mean LRT temperature trends (Fig. 1 second to fifth column) reveals that the trend features are generally robust across all seasons, with SH spring (SON) and winter (JJA) demonstrating a similar trend pattern compared to the overall pattern.

In the tropics, the increase in LRT temperature is most pronounced in the NH winter season (DJF) with up to 0.6 K/decade and persists in all other seasons. This is consistent with results of Zolghadrshojaee et al. (2024) over a similar time period, showing cold-point temperature trends in the tropics. The hemispheric asymmetry, characterized by larger trends in the SH subtropics, is predominantly attributed to SON and to some extend to NH summer season (JJA), exhibiting significant trends

(> 0.5 K/decade) at approximately 30° S. The largest trends in the SH polar regions can be found in SON and JJA with a LRT temperature decrease of up to 1 K/decade, but they are significant only in JJA. In the NH, overall no significant LRT temperature change is found.

Analogous to LRT temperature, for LRT height, the hemispheric asymmetry around the equator with negative trends in the SH extratropics is predominantly attributed to the SON and JJA seasons, although trend values are not significant. A substantial increase in the LRT height of SH polar regions can also be discerned during the SON season, albeit with considerable uncertainties. Highly significant positive LRT height trends in the SH polar region during the SH winter (JJA) are striking, exceeding 200 m/decade. However, the LRT is less well defined in the extremely cold SH polar winter, which could lead to unreasonably high LRT values (Zängl and Hoinka, 2001). This decreases our confidence in these trend values. The NH spring season (MAM) is characterized by the least pronounced trends in LRT height, with virtually no significant trends.

The observed seasonal patterns in LRT temperature and height trends could be partly related to changes in the BDC. The BDC is the primary mechanism through which ozone is transported in the stratosphere, moving it from the tropics towards the poles. The BDC exhibits pronounced seasonality (Butchart, 2014), and is strongest in hemispheric winter and spring time. Recent studies have shown that the BDC has likely decelerated in the 21<sup>st</sup> century, which is linked to ozone recovery in model studies (Polvani et al., 2018). This effect is stronger in the SH, leading to a hemispheric asymmetry in the BDC trends, with a weakening BDC in the SH relative to the NH (Ploeger and Garny, 2022; Sweeney et al., 2025). This also affects stratospheric temperature, warming the SH subtropical lower stratosphere while cooling the Antarctic lower stratosphere (Ladstädter et al., 2023; Sweeney et al., 2025). This is consistent with the observed seasonal LRT temperature trends, showing a dipole structure between the SH subtropics and the SH polar region specifically in the JJA and SON seasons, and, to a lesser extent, also in the DJF and MAM seasons. Therefore, the warming in the tropics and cooling in high latitudes in winter and spring could be a signature of the observed weakening of the BDC (Khaykin et al., 2017; Fu et al., 2019; Ladstädter et al., 2023; Sweeney et al., 2025).

We now turn to zonally and meridionally resolved LRT temperature and height trends in Fig. 2. While a significant increase of LRT temperature is found in most longitudes in the tropics, LRT height trends are not homogeneous across longitudes. Furthermore, Fig. 2 indicates that the peaks in the zonal mean LRT temperature in the SH subtropics (Fig. 1) are primarily attributable to changes over the South Pacific region, in conjunction with generally larger trends in the SH subtropics of up to 0.6 K/decade. Positive LRT height trends are particularly pronounced in the SH polar region and over Asia, corresponding to negative (though mostly not significant) LRT temperature trends in these areas.

The spatial distribution of LRT temperature and height trends divided by season (Fig. 3) provides further detail. In the SH Pacific region, positive LRT temperature trends are largest in Austral spring and summer (SON, DJF) and persists in the other seasons (MAM, JJA). Negative LRT temperature trends are most pronounced over Asia and the Middle East in boreal winter (DJF) and over the SH polar region in Austral winter (JJA).

There are pronounced negative LRT height trends in the SH Pacific region peaking during both austral summer and winter (DJF, JJA), amounting to a decrease of more than 100 m/decade. Concurrently, positive LRT height trends over the SH polar region during austral winter (JJA) correspond to negative LRT temperature trends in that region. Over Asia and the Middle

Figure 2. Observed trends in lapse rate temperature (left) and height (right), from 2002 to 2024, in  $10^{\circ}$  x $10^{\circ}$  horizontal resolution. Stippled areas are regions which are not significant at the 95% level.

East, positive LRT height trends coincide with negative LRT temperature trends in boreal winter (DJF). Additionally, positive LRT height trends extend across a wide range of longitudes during boreal fall and summer (SON, JJA) in the NH midlatitudes. The prominent changes observed in large parts of the subtropics can be related to the location of the "tropopause break", where the tropopause exhibits a strong discontinuity in height, transitioning from the high tropical to the lower extratropical regime (Randel et al., 2007). Therefore, a positive height trend in this region could indicate a potential expansion of the tropical regime (Seidel and Randel, 2007; Xian and Homeyer, 2019; Weyland et al., 2025).

The pronounced positive height trends over land areas in the NH midlatitudes, especially during boreal winter (DJF) (Fig. 3, right panels), are consistent with the troposphere warming more over land compared than over the ocean due to differences in heat capacity (Santer et al., 2018). Such tropospheric warming is expected to manifest as an increase in the height of the tropopause.

Finally, the monthly time series of LRT temperature and height are presented to facilitate a better understanding of the relation between the observed trends and the variability of LRT height and temperature.

Figure 4 illustrates the LRT temperature and height trends from 2002 to a specified end date (upper panels) and the associated anomalies (lower panels) over time, across all latitudes. The minimum trend period shown corresponds to a 10-years span, from 2002 to 2012.

The upper panels of Fig. 4(a)(b) indicate the time of emergence of significant LRT trends in the presence of the variability shown in the lower panels. Figure 4(a) (top panel) shows that the LRT temperature trends in the tropics and subtropics become statistically significant around 2020 and are also increasing. This is not only due to the extended trend period, but also to the exceptionally strong anomalies in LRT temperature after 2020 (Fig. 4(a), bottom panel). In the Southern and Northern high latitudes, the LRT temperature trends are not significant, although they are large. This is mainly due to the high variability especially during the SON season in the Southern high latitudes, and during DJF and MAM seasons in the Northern high

**Figure 3.** Observed seasonal trends in lapse rate temperature (left) and height (right), from 2002 to 2024, in  $10^{\circ}$  x  $10^{\circ}$  horizontal resolution. From top to bottom: DJF, MAM, JJA, and SON. Stippled areas are regions which are not significant at the 95% level.

**Figure 4.** Time series of lapse rate tropopause (a) temperature and (b) height. The respective upper panels show the trend values from the beginning of the time series (2002) up to the given date, showing the convergence of the trend over time. The respective lower panels show the observed anomalies, with ENSO and QBO removed.

latitudes. In contrast, for the LRT height trend (Fig. 4(b), upper panel), a significant height increase is observed in the mid to high latitudes of the Northern Hemisphere from around 2016. In the Southern high latitudes, the LRT height trend becomes significant from about 2022.

The anomaly time series shown in the lower panels of Fig. 4(a) and (b) have ENSO and QBO removed, as described by their respective proxies (see Sect. 2). This reduces variability in the tropics and subtropics to a certain extend, while trend values change only slightly (not shown). However, substantial variability remains, with the period from 2020 to 2023 standing out as particularly noteworthy. The most pronounced variability is observed in the tropics and the high latitudes of the Southern Hemisphere. During the SON season between 2020 and 2023, the Southern high latitudes exhibit strong negative anomalies in tropopause temperature and positive anomalies in tropopause height, likely linked to strong SH polar vortices.

The substantial post-2020 tropopause temperature and height anomalies emerge not only in the high Southern latitudes but also in the tropical and subtropical regions of both hemispheres. These tropical anomalies contrast with the patterns seen in the high Southern latitudes. Studies like Yulaeva et al. (1994) suggest a dynamical connection between high and low latitudes in the lowermost stratosphere, leading to anti-correlated temperatures. This may also include the tropopause region and help explain the observed anticorrelation in tropopause temperature and height between these regions.

# 4 Conclusions

This study exploits the power of vertically high-resolution GNSS RO satellite observations to provide seasonally and regionally detailed trends in the LRT. The analysis of the LRT temperature and height reveals that there have been statistically significant changes in the structure of the global tropopause over the past 23 years. The tropical tropopause has warmed, with implications for the stratospheric water vapor concentration and global climate. The warming was particularly strong in the SON and DJF seasons with up to 0.6 K/decade, with larger values over the South Pacific.

In contrast to temperature, LRT height did not change significantly in the tropics, except in the NH summer where it increased by about 90 m/decade. This appears to be linked to the anomalous warming of the lowermost stratosphere in the tropical and subtropical SH, counteracting the LRT height increase induced by a warmer troposphere.

Outside the tropics, a discernible change in LRT temperature has only been observed in the SH polar regions during the SH winter, with a decrease of up to 1 K/decade. In contrast, LRT height has increased significantly at almost all latitudes outside the tropics, although with strong seasonal and regional variations. In the South Pacific region, areas of LRT warming partly correspond to a decrease in LRT height of more than 100 m/decade. The SH polar winter exhibits the largest positive height trends with more than 200 m/decade, but with a less well defined LRT definition. In the NH, the main contributions to the significant LRT height increase of 50 m/decade to 100 m/decade come from the summer and autumn seasons, but large height trends are also observed in winter over parts of the Asian and the Middle Eastern land masses.

Significant LRT temperature trends emerge from the background variability around 2020 in the tropics and subtropics. LRT height trends become significant as early as 2016, but only in the NH, and in high Southern latitudes around 2022. The results indicate a change in the global atmospheric climate system and the global circulation patterns over the last decades, monitored by sensitive evidence of tropopause changes.

Data availability. The GNSS RO profiles can be acquired from the ROM SAF (ROM SAF, 2021).

The tropopause data used in this study are available here: https://doi.org/10.5281/zenodo.15313075.

The Singapore wind data was downloaded from https://acd-ext.gsfc.nasa.gov/Data\_services/met/qbo/QBO\_Singapore\_Uvals\_GSFC.txt, and the ENSO index from https://www.cpc.ncep.noaa.gov/data/indices/ersst5.nino.mth.91-20.ascii.

*Author contributions*. Conceptualization: FL, MS, SS, and AS; Software: FL; Formal analysis: FL; Visualization: FL; Investigation: FL, MS, SS; Writing – original draft: FL; Writing – review and editing: MS, SS, AS; Supervision: AS.

Competing interests. The authors declare that they have no conflict of interest.

Acknowledgements. We thank Hallgeir Wilhelmsen for implementing the tropopause algorithm and helpful discussions. The authors acknowledge the financial support by the University of Graz.

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
