# Peer review of "Observed changes in the temperature and height of the globally resolved lapserate tropopause"

_EGUsphere, 2025_

## Referee Comment (RC1)

Review of

**Observed changes in the temperature and height of the globally resolved lapserate tropopause**

by Ladstädter et al.

This study provides an update of the lapse-rate tropopause trends based on GNSS-RO data. Tropopause height and temperature trends are provided globally with a focus on seasonal and regional features. The direct comparison of long-term changes of tropopause height and temperature allows a better understanding of the coupling of these quantities. The manuscript is well written and clearly structured. I recommend publication after the following comments have been addressed.

**Major comments:**

1. The results are clearly presented; however, detailed discussions of potential drivers of the trends are missing. There are a few hints on possible mechanisms, but these remain highly speculative. To give one example in line 152-153 a possible connection between tropopause changes in the subtropics and tropical widening is mentioned. What is the mechanism of this potential connection? Would it also impact tropical tropopause trends? Here and in other places, please add more detailed discussions as this would greatly benefit the manuscript.

2. Connected to the point above, please clarify the potential role of BDC transport changes for tropopause trends. How could BDC changes impact the tropopause in the tropics and at mid- to high latitudes? Are the observed tropopause trends consistent with BDC changes derived from observations? Line 135 hints at such a consistency, but ignores the hemispheric asymmetry in observed BDC changes derived from trace gas observations and also found in reanalyses. Is all of this consistent with model results?

3. Discuss if and how sampling inhomogeneities between the earlier GNSS-RO missions such as CHAMP and GRACE and later missions such as COSMIC can impact trend estimates especially for the regional and seasonal tropopause trends given in the paper.

4. Provide some discussions of the variability in tropopause height and temperature explained by the regression proxies QBO and ENSO. Why is no proxy for stratospheric aerosol included here? Given that stratospheric aerosol can impact lower stratospheric temperatures, it can potentially also play a role for tropopause temperature and height. Please also provide a discussion on if and how these proxies impact the trend estimates.

**Minor comments**

Section 2, first paragraph: The authors explain how water vapor is negligible for calculating the temperature profiles. Is this true under all circumstances, i.e., also after the Hunga Tonga eruption? Also, it would be nice to give the time period over which the data is available and analyzed in this first paragraph.

Line 87: Why is the climatology over 2007 to 2023 subtracted from the monthly mean fields and not the full climatology?

Line 129: Please provide more information on the fact that the LRT is less well defined in the cold SH polar regions. How could this impact your analysis?

Line 170: Would it be fair to say starting in 2016?

---

## Author Comment (AC1)

**Response to reviewer 1 (RC1)**

*This study provides an update of the lapse-rate tropopause trends based on GNSS-RO data. Tropopause height and temperature trends are provided globally with a focus on seasonal and regional features. The direct comparison of long-term changes of tropopause height and temperature allows a better understanding of the coupling of these quantities. The manuscript is well written and clearly structured. I recommend publication after the following comments have been addressed.*

Thank you for your constructive feedback and your recommendation for publication. Please find our responses to your comments below.

**Major comments:**

*1. The results are clearly presented; however, detailed discussions of potential drivers of the trends are missing. There are a few hints on possible mechanisms, but these remain highly speculative. To give one example in line 152-153 a possible connection between tropopause changes in the subtropics and tropical widening is mentioned. What is the mechanism of this potential connection? Would it also impact tropical tropopause trends? Here and in other places, please add more detailed discussions as this would greatly benefit the manuscript.*

Thank you for this important comment. We agree that some of the presented discussions about possible drivers are too concise. We have added more details about tropical widening in line 171-174. We have also added a short discussion about a possible driver of large tropopause height trends over land in line 175-178. We then extended the comment about the role of BDC changes, please see the response below.

We also agree that it would be interesting to know more about the potential drivers of the observed tropopause changes. However, a more detailed analysis of these drivers would require modeling efforts, which are beyond the scope of this work. Here, our focus is on presenting the observed changes, which could inform future modeling efforts.

We changed line 171 to:

The prominent changes observed in large parts of the subtropics can be related to the location of the "tropopause break", where the tropopause exhibits a strong discontinuity in height, transitioning from the high tropical to the lower extratropical regime (Randel et al., 2007). Therefore, a positive height trend in this region could indicate a potential expansion of the tropical regime (Seidel and Randel, 2007; Xian and Homeyer, 2019; Weyland et al., 2025).

We added line 175:

The pronounced positive height trends over land areas in the NH midlatitudes, especially during boreal winter (DJF) (Fig. 3, right panels), are consistent with the troposphere warming more over land compared than over the ocean due to differences in heat capacity (Santer et al., 2018). Such

tropospheric warming is expected to manifest as an increase in the height of the tropopause.

*2. Connected to the point above, please clarify the potential role of BDC transport changes for tropopause trends. How could BDC changes impact the tropopause in the tropics and at mid- to high latitudes? Are the observed tropopause trends consistent with BDC changes derived from observations? Line 135 hints at such a consistency, but ignores the hemispheric asymmetry in observed BDC changes derived from trace gas observations and also found in reanalyses. Is all of this consistent with model results?*

We have augmented the paragraph about the potential connection to the BDC in line 144 with more details and additional citations, and hope that this clarifies the role of the BDC. However, again we want to clarify that the main purpose of this paper is to show observed trends, and not to attribute the drivers of the trends, which would require a different study design including modeling.

The observed seasonal patterns in LRT temperature and height trends could be partly related to changes in the BDC. The BDC is the primary mechanism through which ozone is transported in the stratosphere, moving it from the tropics towards the poles. The BDC exhibits pronounced seasonality (Butchart, 2014), and is strongest in hemispheric winter and spring time. Recent studies have shown that the BDC has likely decelerated in the 21st century, which is linked to ozone recovery in model studies (Polvani et al., 2018). This effect is stronger in the SH, leading to a hemispheric asymmetry in the BDC trends, with a weakening BDC in the SH relative to the NH (Ploeger and Garny, 2022; Sweeney et al., 2025). This also affects stratospheric temperature, warming the SH subtropical lower stratosphere while cooling the Antarctic lower stratosphere (Ladstädter et al., 2023; Sweeney et al., 2025). This is consistent with the observed seasonal LRT temperature trends, showing a dipole structure between the SH subtropics and the SH polar region specifically in the JJA and SON seasons, and, to a lesser extent, also in the DJF and MAM seasons. Therefore, the warming in the tropics and cooling in high latitudes in winter and spring could be a signature of the observed weakening of the BDC (Khaykin et al., 2017; Fu et al., 2019; Ladstädter et al., 2023; Sweeney et al., 2025).

*3. Discuss if and how sampling inhomogeneities between the earlier GNSS-RO missions such as CHAMP and GRACE and later missions such as COSMIC can impact trend estimates especially for the regional and seasonal tropopause trends given in the paper.*

It is correct that the transition from the early CHAMP period which had fewer occultation events (2002 to mid-2006), to the later COSMIC/METOP period (mid-2006 to 2024) requires careful consideration of its impact on climatological time series. A widely used approach here is to use a reference field, such as the ERA5 reanalysis to estimate the bias due to incomplete sampling. However, for the following reasons, in this study we decided to use the data directly, without sampling bias correction: Firstly, RO has a considerably better vertical resolution than ERA5, particularly around the tropopause. Therefore, we expect ERA5 to miss details about the lapse rate structure and conclude that it is not a valid reference for this application. Secondly, as the timeseries now extends to 2024, the fraction of the timeseries with less dense sampling compared to the later period has become small (4.5 yrs compared

to 18.5 yrs), and the trend uncertainty due to sparse sampling has decreased accordingly. Finally, we nevertheless redid our analysis with sampling bias correction based on ERA5 fields, and the overall trend structure did not change (Fig. R1). The sampling bias correction reduces some noise in the regionally resolved patterns, but does not change our conclusions.

[Figure]

Fig. R1: Observed seasonal trends in lapse rate height (top rows) and temperature (bottom rows). The top panel shows the trends without sampling bias correction, the bottom panel with sampling bias correction using ERA5 reference fields.

In accordance with our aim of being as close as possible to the original measurements, we therefore chose not to apply sampling bias correction here. We have added the following paragraph to line 94-100 of the manuscript to explicitly discuss this topic:

We note that the number of measurements involved in the RO time series varies due to the transition from the early single-mission RO period (2002 to mid-2006) with less measurements to the later period (mid-2006 to 2024) when more RO missions contributed (Gleisner et al., 2020). In many applications, it is favorable to apply a correction for the resulting sampling bias caused by incomplete atmospheric sampling using a background field such as the ERA5 reanalysis (Foelsche et al., 2008; Scherllin-

Pirscher et al., 2011). In our case, applying such a correction does not alter the overall trend values and patterns (not shown). Therefore, we have chosen not to correct for these sampling inhomogeneities here, since the decreased vertical resolution of such a background field could tamper with the pure observational information contained in the RO profiles.

*4. Provide some discussions of the variability in tropopause height and temperature explained by the regression proxies QBO and ENSO. Why is no proxy for stratospheric aerosol included here? Given that stratospheric aerosol can impact lower stratospheric temperatures, it can potentially also play a role for tropopause temperature and height. Please also provide a discussion on if and how these proxies impact the trend estimates.*

In our analysis we removed known modes of internal climate variability (QBO, ENSO) while retaining externally forced signals, including variability related to stratospheric water vapor and aerosols. QBO and ENSO explain a notable fraction of the variability in tropopause temperature and height, particularly in the tropics and subtropics. While we acknowledge that aerosols potentially influence tropopause parameters and trends, too, we want to retain such external forcings in our analysis, in line with our objective of showing the observed changes in tropopause parameters rather than analyzing the drivers of these changes.

The two proxies of internal variability used in this work (QBO and ENSO) mainly reduce the variability of the anomalies time series (Fig. R2), and with that the standard error of the trend. With this, more regions become statistically significant (Fig. R3). The overall trend values and patterns change only slightly, and do not change our conclusions.

We have adapted a paragraph in line 193-197 in the results chapter to discuss the impact of the proxies, in addition to the existing paragraph in the Methods section:

The anomaly time series shown in the lower panels of Fig. 4(a) and (b) have ENSO and QBO removed, as described by their respective proxies (see Sect. 2). This reduces variability in the tropics and subtropics to a certain extend, while trend values change only slightly (not shown). However, substantial variability remains, with the period from 2020 to 2023 standing out as particularly noteworthy. The most pronounced variability is observed in the tropics and the high latitudes of the Southern Hemisphere.

[Figure]

Fig. R2: The observed anomalies of the LRT height (top panel) and LRT temperature (bottom panel). The top rows have ENSO and QBO removed, the bottom rows do not.

[Figure]

Fig. R3: The observed seasonal trends without considering QBO and ENSO (top panel) and with considering QBO and ENSO (bottom panel). The top rows show LRT height trends, the bottom rows show LRT temperature trends.

**Minor comments:**

*Section 2, first paragraph: The authors explain how water vapor is negligible for calculating the temperature profiles. Is this true under all circumstances, i.e., also after the Hunga Tonga eruption*

We agree that water vapor can affect the retrieved dry temperature profiles. After the Hunga Tonga eruption, extremely high water vapor concentrations were injected into the stratosphere, but mainly at altitudes well above the tropopause. In these regions, the refractivity retrieval can indeed be disturbed, leading to a "fictional" cooling signal in the dry temperature profile until approximately early February. However, the water vapor plume dispersed and diffused rapidly, and the direct effect on the RO dry temperature retrieval became negligible thereafter. This interpretation is supported by the close agreement between RO and independent MLS temperature profiles after the initial weeks as shown by Stocker et al., 2024 (doi: 10.1038/s43247-024-01620-3).

*Also, it would be nice to give the time period over which the data is available and analyzed in this first paragraph.*

Added.

*Line 87: Why is the climatology over 2007 to 2023 subtracted from the monthly mean fields and not the full climatology?*

We commonly use this time period in our analyses and are using it here to simplify potential internal comparisons. There is no specific reason for this time period. In our experience, using the full climatology would only change the results to a small extent.

*Line 129: Please provide more information on the fact that the LRT is less well defined in the cold SH polar regions. How could this impact your analysis?*

We changed this line to better describe this situation, and added a citation to line 141-142:

However, the LRT is less well defined in the extremely cold SH polar winter, which could lead to unreasonably high LRT values (Zängl and Hoinka, 2001). This decreases our confidence in these trend values.

*Line 170: Would it be fair to say starting in 2016?*

When looking at the SH LRT temperature anomalies, SON seasons, from 2016 to 2023, we also observe several positive anomalies, in 2016, 2017, and 2019. We therefore prefer to keep the sentence as it is, saying that strong negative anomalies are persistently observed from 2020 to 2023.

---

## Author Comment (AC2)

**Response to reviewer 2 (RC2)**

*The study by Ladstaedter et al. shows Global Navigation satellite based temperature analysis of the tropopause altitude and temperature. The analysis covers 2002-2024 and provides a global view on the tropopause using the lapse rate tropopause LRT according to WMO. The authors further apply multiple linear regression with regressors for QBO and ENSO to derive trends of the LRT-height, and the LRT temperature. Seasonally and regionally resolved trends are presented. The authors also check for trend robustness with regard to the end date of data set.*

*They find a significant positive temperature increase at the LRT in the tropics and southern subtropics over time with, but no significant rise in LRT altitude. Instead the northern extratropics show an increase in LRT-height, and partly weak positive trend.*

*Despite providing no analysis for the trend differences the paper provides extremely interesting data to the community. It is very well written, Figures are clear and the topic is well within the scope of ACP.*

*I recommend the paper for publication with just very few minor remarks to be considered and which are given below.*

Thanks to the reviewer for your recommendation to publish the manuscript. We provide responses to your comments below.

***Data:***

*1) Given the change of available profiles from COSMIC in the year 2007: How does this affect the trend estimates and statistics?*

It is correct that the transition from the early CHAMP period which had fewer occultation events (2002 to mid-2006), to the later COSMIC/METOP period (mid-2006 to 2024) requires careful consideration of its impact on climatological time series. A widely used approach here is to use a reference field, such as the ERA5 reanalysis to estimate the bias due to incomplete sampling. However, for the following reasons, in this study we decided to use the data directly, without sampling bias correction: Firstly, RO has a considerably better vertical resolution than ERA5, particularly around the tropopause. Therefore, we expect ERA5 to miss details about the lapse rate structure and conclude that it is not a valid reference for this application. Secondly, as the timeseries now extends to 2024, the fraction of the timeseries with less dense sampling compared to the later period has become small (4.5 yrs compared to 18.5 yrs), and the trend uncertainty due to sparse sampling has decreased accordingly. Finally, we nevertheless redid our analysis with sampling bias correction based on ERA5 fields, and the overall trend structure did not change (Fig. R1). The sampling bias correction reduces some noise in the regionally resolved patterns, but does not change our conclusions.

[Figure]

Fig. R1: Observed seasonal trends in lapse rate height (top rows) and temperature (bottom rows). The top panel shows the trends without sampling bias correction, the bottom panel with sampling bias correction using ERA5 reference fields.

In accordance with our aim of being as close as possible to the original measurements, we therefore chose not to apply sampling bias correction here. We have added the following paragraph to line 94-100 of the manuscript to explicitly discuss this topic:

We note that the number of measurements involved in the RO time series varies due to the transition from the early single-mission RO period (2002 to mid-2006) with less measurements to the later period (mid-2006 to 2024) when more RO missions contributed (Gleisner et al., 2020). In many applications, it is favorable to apply a correction for the resulting sampling bias caused by incomplete atmospheric sampling using a background field such as the ERA5 reanalysis (Foelsche et al., 2008; Scherllin-Pirscher et al., 2011). In our case, applying such a correction does not alter the overall trend values and patterns (not shown). Therefore, we have chosen not to correct for these sampling inhomogeneities here, since the decreased vertical resolution of such a background field could tamper with the pure observational information contained in the RO profiles.

*2) Though the GNSSS data are a valuable and well established data set for deriving tropopause altitudes, the authors should provide in this manuscript a short paragraph on temperature uncertainties since they derive temperature trends here, despite given in other literature sources.*

We added the following in the data section (line 71-74) to discuss the RO uncertainties in some more detail, and to cite some more references:

The systematic uncertainty in the UTLS is estimated to be below 0.1 K, with the lowest values around the tropopause (Scherllin-Pirscher et al., 2011b, 2021; Schwarz et al., 2017). Random uncertainty can already be neglected for small aggregation sizes, and the structural uncertainty due to processing choices is less than 0.1 K in that region (Steiner et al., 2020b).

*Methods:*

*Regarding potential multiple tropopauses: How are these treated?*

*Does the analysis take the upper, or the lower one, or exclude such multiple cases particularly in the subtropics?*

We only consider the first (lowest) tropopause found by the lapse rate algorithm. Analyzing the more complex vertical structure of the subtropics, which often manifests as multiple tropopauses, is beyond the scope of this work. However, this has been analyzed using RO, e.g. by Wilhelmsen et al. 2020 (doi: 10.1029/2020GL089027). To clarify in the manuscript that we use only the first tropopause, we have added the following to line 87-89:

According to the WMO lapse rate definition, a second tropopause might occur above the first, which is a phenomenon common in the extratropics (e.g., Randel et al., 2007). In this study, however, only the lowest (first) tropopause is considered when multiple tropopauses are found in the temperature profile.

---

## Referee Report (RR1)

**Second review of**

**Observed changes in the temperature and height of the globally resolved lapserate tropopause**

by Ladstädter et al.

Overall, the authors have adequately responded to my concerns given in the first review. Most importantly, in the revised version of the manuscript, the authors have 1) added a discussions of possible drivers of the observed changes and 2) provided evidence that a sampling bias correction does not change the conclusions of the study.